# Deep Learning Prediction of Pathologic Complete Response in Breast Cancer Using MRI and Other Clinical Data: A Systematic Review

Nabeeha Khan [†], Richard Adam [†], Pauline Huang, Takouhie Maldjian and Tim Q. Duong *

Department of Radiology, Montefiore Medical Center, Albert Einstein College of Medicine, Bronx, New York, NY 10461, USA
* Correspondence: tim.duong@einsteinmed.edu
† These authors contributed equally to this work.

**Abstract:** Breast cancer patients who have pathological complete response (pCR) to neoadjuvant chemotherapy (NAC) are more likely to have better clinical outcomes. The ability to predict which patient will respond to NAC early in the treatment course is important because it could help to minimize unnecessary toxic NAC and to modify regimens mid-treatment to achieve better efficacy. Machine learning (ML) is increasingly being used in radiology and medicine because it can identify relationships amongst complex data elements to inform outcomes without the need to specify such relationships a priori. One of the most popular deep learning methods that applies to medical images is the Convolutional Neural Networks (CNN). In contrast to supervised ML, deep learning CNN can operate on the whole images without requiring radiologists to manually contour the tumor on images. Although there have been many review papers on supervised ML prediction of pCR, review papers on deep learning prediction of pCR are sparse. Deep learning CNN could also incorporate multiple image types, clinical data such as demographics and molecular subtypes, as well as data from multiple treatment time points to predict pCR. The goal of this study is to perform a systematic review of deep learning methods that use whole-breast MRI images without annotation or tumor segmentation to predict pCR in breast cancer.

**Keywords:** neoadjuvant chemotherapy; convolutional neural networks; machine learning; artificial intelligence; molecular subtypes; magnetic resonance imaging; dynamic contrast-enhanced MRI

## 1. Introduction

Neoadjuvant chemotherapy (NAC) [1] is commonly prescribed to reduce tumor burden prior to breast cancer surgical management, in order to improve surgical outcomes and potentially allow breast conservation in otherwise unresectable tumors or tumors requiring mastectomy [2]. Pathological complete response (pCR), defined as the absence of any residual invasive disease, is often used to assess NAC response via pathological analysis of biopsied or dissected tissue at the end of the NAC treatment course [3]. Patients with pCR are more likely to be candidates for breast-conserving surgery and are also likely to have longer progression-free survival and overall survival [3]. Thus, the ability to longitudinally monitor individual response to NAC and to determine patient's likelihood to respond to NAC early in the treatment course is clinically important because it could help to minimize unnecessary toxic NAC and modify regimens mid-treatment to achieve better efficacy. A major challenge to date is the lack of reliable methods to assess efficacy early in the NAC course. Radiological prediction of pCR is a desirable alternative to pathology because it is non-invasive, and able to assess the entire breast at once without being limited to the biopsied area.

Contrast-enhanced MRI (CE-MRI), in which images are acquired serially following a bolus injection of a contrast agent, provides important dynamic vascular and physiological

information about the tumor that could be useful for predicting response to NAC [4]. Radiographic complete response based on CE-MRI has been shown to have high sensitivity and specificity to predicting pCR [5]. Response to therapy has proven to be a strong predictor of outcome, with patients who achieve pCR demonstrating improved survival compared with non-pCR patients While DCE is frequently used for this analysis, other pulse sequences such as T2 weighted images, and diffusion-weighted images can also be used to predict pCR.

Machine learning (ML) is increasingly being used in radiology and medicine [6–8]. In contrast to conventional analysis methods which need to specify the relationships amongst data elements to outcomes, ML employs computer algorithms to identify relationships amongst different data elements to inform outcomes without the need to specify such relationships a priori. ML can outperform human experts in performing many tasks in medicine [9]. In addition to approximating physician skills, ML can also detect novel relationships not readily apparent to human perception, especially in large complex datasets. There are multiple types of learning methods (supervised, self-supervised, and unsupervised), each having its own advantages and disadvantages. Supervised learning learns from labeled data to perform tasks such as prediction and classification of data. A comparatively small data set is needed but the disadvantage is that labeled datasets for a specific task needs to be provided. Self-supervised learning exploits unlabeled data to yield labels. This eliminates the need for manually labeling data, which is a tedious process, but larger datasets are needed. Supervisory signals provide feedback to train the network, without requiring a labeled dataset. Unsupervised learning needs no corresponding classification or label, and the algorithm finds underlying patterns with each dataset. The disadvantage is that is it requires large datasets. One of the most popular unsupervised learning, deep learning (DL) methods [10] used on images is the Convolutional Neural Networks (CNN) [11]. CNN is well suited for image recognition and tasks that involve the processing of pixel data. CNN is particularly suitable for computer vision tasks and for applications where object recognition such as facial recognition and radiological images. CNN methods do not require radiologists to contour the tumor on images.

In addition to imaging data, inclusion of other clinical data (such as demographic, race and ethnicity, molecular subtypes, and laboratory tests) could improve pCR prediction [12]. Molecular subtypes could significantly affect pCR. For example, HER2-positive (elaborating excess human epidermal growth factor receptor-2 protein) and triple-negative tumors achieved significantly higher rates of pCR compared to luminal A subtype tumors, which can aid physicians in creating treatment plans [12]. Hormone receptor positive (HR+) (breast cancer cells with estrogen and/or progesterone receptors) and HER2-positive cancers have receptors that can be targeted, allowing for directed therapy which can significantly improve prognosis. Inclusion of specific molecular subtypes could enhance the ability to predict pCR. Epigenetic factors should also be considered, as molecular status can change as treatment progresses, and MRI can prove to be valuable in monitoring changes over multiple timepoints. Young age has a negative prognosis and puts this cohort at increased risk of dying. The application of deep learning, with its ability to manage large and multiple complex datasets (including imaging and non-imaging data), holds promise of being able to accurately predict pCR to guide breast cancer management.

There are many review papers on using ML analysis of MRI data to predict pCR in breast cancer (see reviews [13,14]) but most were of supervised ML, such as support vector machine, random forest, decision tree, extreme gradient boosting, Boosted tree, Bayesian methods, among others (see reviews [13,14]). Supervised ML methods use extracted image features (i.e., tumor volume, diameter, radiomic features), that usually require expert radiologists to contour the tumors and/or identify features to train the ML algorithms. By contrast, reviews that focus on DL methods (such as CNN) in which whole breast MRI images are used without manual contouring or annotation of the tumor to predict pCR is limited [13–15].

We thus took on a review that focuses on DL which allows whole-breast MRI images without annotation or tumor segmentation (i.e., feature agnostic DL approach) to predict pCR in breast cancer. Our review differed from prior related reviews in that we compared image types used (i.e., DCE or post contrast), whether pre-training, transfer learning or data augmentation was used, whether CNN models include molecular subtypes, multiple treatment time points, multi-institutional data, as well as whether saliency maps (heatmaps) were provided. We also compared performance metrics across different DL studies that predict pCR.

## 2. Materials and Methods

No ethics committee approval was required for this systematic review. With the use of Pubmed and Google Scholar, a search of literature was carried out to identify journal articles that presented deep learning techniques in breast MRI to show pCR. To narrow down the search, key words were utilized to accumulate eligible articles. The search terms used were a combination of: "breast MRI", "deep learning" and "pCR (pathological complete response)".

The search of journal articles was narrowed down to original journal articles from 2015 to 1 November 2022. Only articles written in English and including an abstract were selected. The initial screening for eligibility was done by an independent researcher. Initially, articles that included deep learning and breast MRI were included. For this review, the topic was limited to deep learning with MRI and its applications to prediction of pCR post NAC. Segmentation of tumors, while possible with deep learning and found in our search, was not included in this review. We included papers that used ROI notation to assist prior to applying CNN. The PRISMA flowchart is shown in Figure 1.

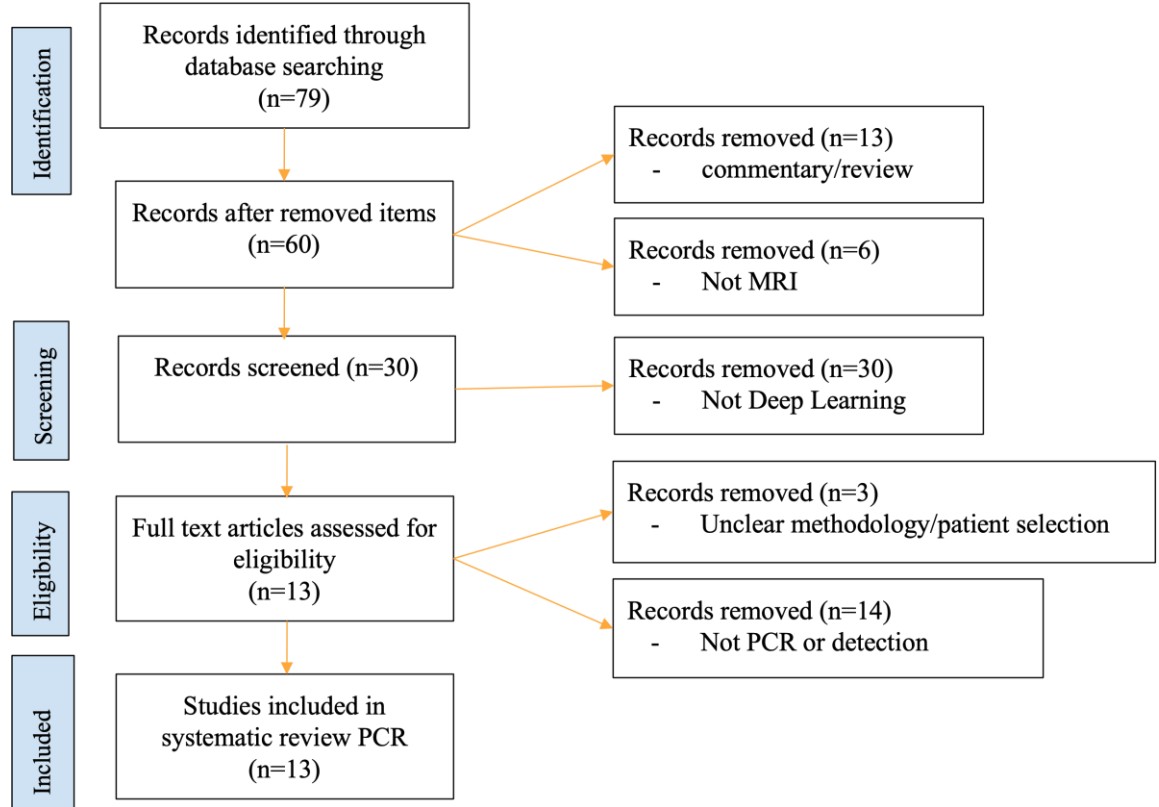

**Figure 1.** PRISMA flowchart. Initially, 79 studies were found on PubMed through our keyword search, of which 66 were removed. The final number of papers reviewed was 13.

## 3. Results and Discussion

A typical workflow of the DL algorithms is shown in Figure 2A. The workflow starts with data inputs, followed by data curation, and then training, validation and evaluation of DL methods. Evaluation metrics could include, but not limited to, accuracy, sensitivity, and specificity as well as clinical outcomes. A typical model of how imaging and non-imaging data are incorporated into the CNN workflow is shown in Figure 2B. DCE MRI or a single post-contrast MRI is often used. Some time, multiple treatment time points are included. Images are first put into the CNN. Multiple treatment time point data, if available, are entered as parallel channels on the CNN. In a separate channel, non-imaging data (such as molecular subtypes and demographics) are entered. The multiple channel networks are then concatenated into a fully connected layer. Note that this is only one of the many models of how different data are incorporated into the CNN to predict pCR.

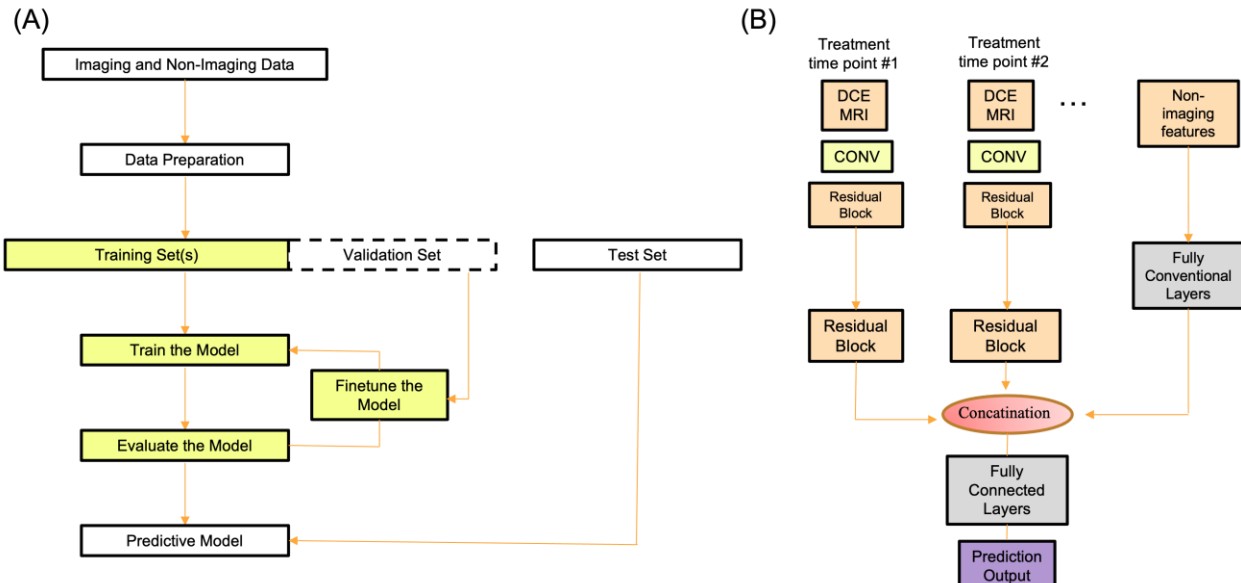

**Figure 2.** (**A**) A typical flowchart of inputs, data curation, training, and evaluation of DL methods. (**B**) A CNN model of how DCE data at multiple treatment time point and non-imaging data are incorporated. Abbreviations: DCE: dynamic contrast enhanced MRI; Conv: convolutional neural network.

### 3.1. CNN Prediction of pCR

Table 1 summarizes the papers on deep learning prediction of pathological complete response in breast cancer using MRI. Braman et al. utilized a multiphasic CNN to predict pCR from 2D DCE MR images acquired pre-NAC. The study focused on patients with HER2+ breast cancer (N = 157) receiving HER2 targeted NAC [16]. They found that prediction models using a combination of pre-contrast and third post contrast MR images showed the highest predictive performance, with an AUC and accuracy of 0.93 and 95%. This study demonstrated the feasibility of DL-based prediction of response including data from multiple sites not included in training. Subtype specific analysis of therapeutic outcome of specific targeted therapies has the potential to precisely guide treatment.

**Table 1.** Summary of papers on deep learning prediction of pathological complete response in breast cancer using MRI.

| Study | Year | Image Type [b] | Pre-Trained or CNN Models | pCR/Non-pCR [c] | Molecular Subtypes | Multiple Time Points | Independ Validation [d] | Multisite | Transfer Learning | Data Augmentation | Heat Maps | AUC [e] | Accu [f] | Sens [f] | Spec [f] |
|---|---|---|---|---|---|---|---|---|---|---|---|---|---|---|---|
| Braman [16] [a] | 2020 | DCE | CNN | 76/81 | no | no | yes | yes | no | yes | no | 0.93 | 86.7% | 75% | 100% |
| Comes [17] | 2021 | CE, T2 | AlexNet | 37/78 | no | no | no | yes | yes | no | no | - | 92.3% | 85.7% | 94.7% |
| Duanmu [18] | 2020 | 3D-CE | VGG13 | 42/112 | yes | no | no | yes | no | no | yes | 0.80 | 83% | 68% | 88% |
| Duanmu [19] | 2022 | DCE, T2 | VGG13 | 42/110 | yes | yes | no | yes | no | no | no | 0.83 ± 0.03 | 81 ± 3% | 68 ± 8% | 86 ± 4% |
| El Adoui [20] | 2019 | CE | CNN | 14/28 | no | no | no | no | no | yes | no | 0.91 | 88% | 92.2% | 79.1% |
| Ha [21] | 2018 | CE | VGG16 | 46/95 | no | no | no | no | no | yes | no | 0.85 | 88 ± 0.6% | 95 ± 3% | 74 ± 5% |
| Huynh [22] | 2017 | DCE | VGGNet | 39/25 | no | no | no | no | yes | no | no | 0.85 ± 0.03 | - | - | - |
| Joo [23] | 2021 | DCE, T2 | ResNet-50 | 133/403 | yes | no | no | no | no | yes | no | 0.888 | - | 66.7% | 93.2% |
| Liu [24] | 2020 | DCE | VGG16 | 40/91 | no | no | no | yes | no | yes | no | 0.72 | 72.5% | 65.5% | 78.9% |
| Massafra [25] | 2022 | CE | AlexNet | 64/161 | no | no | yes | yes | yes | no | no | 0.78 | 77.3% | 71.4% | 80.0% |
| Peng [26] | 2022 | DCE | ResNeXt50 | 83/273 | yes | no | no | no | no | yes | yes | 0.83 | 77.2% | 78.1% | 7.69% |
| Qu [27] | 2020 | DCE | CNN | 132/170 | no | yes [g] | no | no | no | yes | no | 0.97 | - | 96% | 100% |
| Ravichandran [28] | 2018 | DCE | AlexNet | 49/117 | yes | no | no | yes | no | yes | yes | 0.85 | 85% | - | - |

[a] This study tested on HER2+ patients underwent HER2+ targeted NAC only. [b] There are three types of MRI data. DCE: dynamic contrast enhanced MRI (multiple dynamics); CE: first post contrast enhanced MRI; T2: T2-weighted MRI. [c] combined of all data (testing, validation, multiple sites). [d] independent validation was obtained from a separate institution not used in training. [e] performance metrics are given for independent validation data set (if available) or validation data set. [f] pre-NAC and post-NAC MRIs were included by concatenation (not linked) and thus the temporal information might not be optimally utilized. Abbreviations: Independ validation: independent validation; AUC: area under the curve; Accu: Accuracy; Sens: Sensitivity; Spec: Specificity.

Comes et al. [17] reported a transfer learning approach to predict pCR by exploiting, separately or in combination, pre-treatment and early treatment exams. First, low-level features were automatically extracted by a pre-trained CNN overcoming manual feature extraction. Next, an optimal set of most stable features was detected and then used to design an SVM classifier. By combining the optimal features extracted from both pre-treatment and early treatment exams with some clinical features, an accuracy of 92.3%, and an AUC value of 0.90, were returned the independent test, respectively. They concluded that the low-level CNN features have an important role in the early evaluation of the NAC efficacy by predicting pCR.

Duanmu et al. (2020) studied 3D T1-weighted post-contrast whole images and included molecular and demographic data in their analysis [18]. Their CNN model differs from conventional CNNs in that MRI data and non-imaging data are convolved to inform each other through interactions, instead of a concatenation of multiple data type channels. This is achieved by channel-wise multiplication of the intermediate results of imaging and non-imaging data. Using a subset of curated data from the I-SPY-1 TRIAL of 112 patients with stage 2 or 3 breast cancer with breast tumors underwent NAC, they found an accuracy of 0.83, AUC of 0.80, sensitivity of 0.68 and specificity of 0.88. This model significantly outperforms models using imaging data only or traditional concatenation models. Heatmaps of where the algorithms weighted as importance were provided.

Duanmu et al. (2022) used CNN to evaluate 3D DCE whole images at multiple treatment timepoints and incorporated molecular subtype and demographic data [19]. They predicted PCR as well as residual cancer burden (RCB), and progression-free survival (PFS) in breast cancer patients treated with NAC using longitudinal (multiple treatment time points), multiparametric MRI, demographics, and molecular subtypes as inputs. The data came from I-SPY-1 TRIAL (155 patients with stage 2 or 3 breast cancer with breast tumors underwent NAC). The inputs were DCE MRI, and T2-weighted MRI as 3D whole-images without the tumor segmentation, as well as molecular subtypes and demographics. Three ("Integrated", "Stack" and "Concatenation") CNN were evaluated using receiver-operating characteristics and mean absolute errors. The Integrated approach outperformed the "Stack" or "Concatenation" CNN. Inclusion of both MRI and non-MRI data outperformed either alone. The combined pre- and post-neoadjuvant chemotherapy data outperformed either alone. Using the best model and data combination, PCR prediction yielded an accuracy of $0.81 \pm 0.03$ and AUC of $0.83 \pm 0.03$; RCB prediction yielded an accuracy of $0.80 \pm 0.02$ and Cohen's of $0.73 \pm 0.03$; PFS prediction yielded a mean absolute error of $24.6 \pm 0.7$ months (survival ranged from 6.6 to 127.5 months). Deep learning using longitudinal multiparametric MRI, demographics, and molecular subtypes accurately predicts PCR, RCB and PFS in breast cancer patients.

El Adoui et al. applied a 3D CNN to predict pCR from DCE-MRI (N = 42) using two treatment time points [20]. Using a two-branch CNN model to take inputs from MRI pre- and post-chemotherapy, they found an accuracy and ROC AUC of 92.72% and 0.96, respectively in one study and similarly 91.03% and 0.92 in another study. They reported that data augmentation greatly improved prediction performance.

Ha et al. similarly looked at pre-treatment imaging to predict response to NAC [21]. They applied a CNN implemented using the Keras toolbox with tensor flow backend in Python. The CNN architecture followed the general structure of the VGG 16 network. The first post-contrast dynamic T1W images were used prior to NAC for their analysis, producing an 88% overall mean accuracy for 3 class prediction (complete response versus partial response versus progression/no response) of NAC treatment response.

Huynh et al. utilized CNN and Linear Discriminant Analysis (LDA) classifier to predict response to NAC in breast cancer patients (N = 64) using 2D DCE MR images [22]. Features which were first extracted using CNN were then used to train the LDA classifier. They found the best ROC AUC was 0.85 for the pre-contrast DCE. A limitation of this study is that slices containing the tumor were manually selected.

Joo et al. explored the use of deep learning with 3D-CNN when applied to pretreatment MRI (T1W subtraction and T2W images) versus clinical data as well as a multimodal fusion approach using both clinical and pretreatment MRI data in predicting post NAC pCR [23]. They also compared cropped MR images to whole uncropped 3D bilateral images covering the axilla and chest wall. They had the largest cohort for pCR prediction model with breast MRI, with 536 patients with invasive breast cancer. T1W and T2W images showed poorer AUC alone compared to when combined (AUC of 0.725, 0.663, 0.745), clinical data performed better than combined MRI data (AUC of 0.827 versus 0.745), and whole T1W subtraction images performed better than cropped T1W subtraction images (AUC of 0.745 versus 0.624). Using whole images is less labor intensive, allows for multiple findings to be analyzed, and eliminated need for manual or automated segmentation for tumor area extraction. They conjecture that adding multiple post-contrast T1W timepoints allowing use of kinetic information and use of DWI could further improve performance of the model.

Liu et al. used a 12-layer CNN to analyze patients from the I-SPY trial dataset to predict pCR in NAC patients (N = 131) using 2D MR images with first post-contrast DCE [24]. They demonstrated the feasibility of using a CNN algorithm on a multi-institutional MRI dataset, reporting an accuracy, sensitivity, specificity, and ROC AUC of 72.5%, 65.5%, 78.9%, and 0.72, respectively.

Massafra et al. reported the use of DL on different MRI protocols (i.e., axial for private database or sagittal for public database) to predict pCR [25]. By merging the features extracted from baseline MRIs with some pre-treatment clinical variables, accuracies of 84.4% and 77.3% and AUC values of 80.3% and 78.0% were achieved on the independent tests related to the public database and the private database, respectively. AUC values with combined clinical and imaging data exceeded those for either clinical or imaging data alone for both public and private databases.

Peng et al. compared the performances of DL to radiomics analysis in predicting pCR based on pretreatment DCE-MRI in breast cancer [26]. The AUC of the image-molecular radiomics analysis model was 0.755 (95% CI: 0.708, 0.802). The AUC of the image-kinetic-molecular DL model was 0.83 (95% CI: 0.816, 0.847). They concluded that pretreatment DCE-MRI-based DL model is superior to the radiomics analysis model in predicting pCR. Heatmaps of where the algorithms weighted as importance were provided.

Qu et al. applied a 2D CNN to predict pCR from multiple DCE MR images plus molecular subtypes including ER, PR, and HER2 (N = 302) [27]. This model used 12 channels to combine 6 DCE phases from both pre- and post-NAC. They reported an AUC of 0.553 from pre-NAC images and 0.968 from post-NAC images with a combined model of 0.970. A limitation is that pre-NAC and post-NAC MR images were included by concatenation and thus the temporal information might not be optimally utilized.

Ravichandran et al. used a voxel-wise RGB CNN to predict pCR in Pre-NAC patients (N = 166) using selected 2D slices (3 adjacent slices with the largest tumor area) from DCE MR images [28]. The corresponding pre-contrast, first post contrast, and second post contrast images of each slice were placed into red, green, and blue color channels, creating a three-channel color image that is then evaluated on a pixel-wise basis. Inclusion of HER2 status was beneficial, improving the AUC from 0.77 to 0.85. Due to the nature of this approach, heatmaps could be generated and the centers of tumors were found to be the most predictive of pCR. A drawback of this study is that tumor segmentations were provided as part of the dataset and the slices were hand selected. The images from pCR patients were also hyper sampled due to data imbalance. Heatmaps of where the algorithms weighted as importance were provided.

### 3.2. Single Post-Contrast vs. DCE Dynamic Data

Most studies to date use a single post-contrast MRI. Some studies found that, among the multiple DCE dynamics, the first post contrast dynamic to be the most effective at predicting response to therapy and variance reduces with the addition of more contrast

dynamics [22]. Different combinations of contrast images and the incorporation of pre-contrast and the 3rd post-contrast image best predicted response with a highest AUC value of 0.93 [16]. The pre-contrast served the purpose of being a baseline and the comparison between the post-contrast images revealed that the 1st and 2nd postcontrast images take more time for washout patterns to emerge. In general, pre-contrast dynamic is essential for analysis, but the addition of more dynamics is helpful [16,18]. One study showed that CNN predictive model using multiple DCE images was superior to individual dynamics [18]. DL prediction of pCR using multiple DCE dynamics is understudied.

### 3.3. Multiparametric MRI Data

While DCE data are most widely used to predict pCR, other MRI data could also be informative and they include T2-weighted (with and without fat suppression), diffusion imaging, and fat-water imaging. Similarly, background parenchymal enhancement by MRI is informative of cancer risk, recurrence, and outcomes but has not yet been adequately explored for predicting pCR [29].

### 3.4. Data with Multiple Treatment Time Points

CNN model can make use of data at multiple treatment timepoints to improve prediction. Qu et al. showed that using pre-NAC and post-NAC images improved accuracy which in combination with clinical data (receptor status) yielded the best overall performance of all studies, with AUC of 0.97 [27]. Similarly, Duanmu et al. (2020) also employed CNN to evaluate DCE whole images at multiple treatment timepoints and incorporated molecular subtype and demographic data in a component of their analysis [18]. DL prediction of pCR using multiple treatment time points is understudied because it is not trivial to incorporate multiple channels and neural networks in a way that maintain the relationship of multiple treatment time point images and DCE images for the same patients throughout. Other challenges include the risk of overfitting and high computational costs.

### 3.5. Axillary Lymph Nodes

Axillary lymph node status is a metric for pCR and additionally MRI can also provide nodal information non-invasively. A few studies have investigated nodal status using DL predictive models might consider using nodal images that are already part of the patient's MRI exam [30–34]. This is however not without challenge. Axillary lymph nodes are usually small and MR images are usually do not have sufficient contrast and resolution for diagnosis use. However, it may be feasible with improvement in MRI methods and detectors as well as interest in the nodal status using MRI.

In summary, to date, few DL studies include dynamic DCE data and/or multiparametric data, and fewer DL studies include molecular subtypes and/or multiple treatment time points. Few DL prediction of pCR studies make use of axillary lymph node MRI data. Testing using independent validation data set to improve generalizability is rare due to limited data availability. Heatmaps to improve interpretability are also limited. These are possible areas of future investigation.

### 3.6. Current Challenges to Routine Clinical Applications

There are three broad challenges that need to be overcome before DL can have main-stream applications in the clinical settings, namely: generalizability, interpretability, and ethic/legal concerns [30,35]. (a) DL findings need to be broadly generalizable. To improve generalizability, training datasets need not only be large but also diverse to avoid or minimize bias. Publicly available high-quality clinical data with which to test predictive models for pCR are currently limited. To overcome limited dataset, federated learning or collaborative learning can be employed in which a machine learning algorithm is trained on multiple local datasets without data sharing or data exchange across instittuions, thus preserving data privacy and security. Doing so however is not without challenges. (b) DL findings are difficult to interpret because there are many features from which DL draws conclusion and

DL calculations are complex. There are a few things can be done to improve interpretability. Feature importance and accumulated local effects can be assessed. Additional tools such as by evaluating Shapley values can be employed to explain individual outcomes. For imaging applications, heatmaps can be generated to highlight which regions or features on the images the DL algorithm considers to be important for its calculations. (c) There are ethical and legal challenges to overcome. There are uncertainties concerning who bears ultimate responsibility in the case of an incorrect diagnosis. In the case of triaging, we must also determine/decide if it is ethical to deprioritize reading scans from cases ruled "low risk" by a computer model. Transition to mainstream clinical use is also impeded by the fact that DL functions as a "black box" system. Inputs yield outputs without any understanding of the inner workings of how DL arrives at diagnostic predictions, making it challenging to determine whether in any specific case it has made a mistake, and rendering treatment and management decisions based on DL problematic.

### 3.7. How Could DL Be Employed in Practice?

Instead of relying on DL alone, there is also a potential for hybrid intelligence, which combines the expertise of radiologists with deep learning AI. Computer-assisted diagnosis (CAD) systems with deep learning AI can assist radiologists in finding breast cancers and can also potentially increase the radiologist's efficiency. In a more active role, DL can potentially serve as a secondary or concurrent reader. The potential to increase accuracy while decreasing interpretation time/increasing radiologists' productivity is a win-win, as this could help patients and reduce radiologist burn-out. Automated triaging to prioritize scans with findings that require more immediate attention is another potential application of AI. Decision making, such as treatments and patient management, can be guided with DL systems. For example, automated preprocessing, segmentation, detection, and classification of lesions may reduce unnecessary biopsies/surgeries due to the ability of DL to predict the behavior of precancerous lesions. DL systems can be incorporated into decision support systems. The ability to predict patients' treatment responses early on during NAC and potentially alter treatment strategies to optimize outcome allows for individualized treatment and precision medicine. This ability to individualize treatment early on to improve patient survival would be particularly beneficial in underserved areas with lower socioeconomic populations, who currently suffer from worse breast cancer outcomes. ML may thus help to address some current healthcare disparities. Pooling large quantities of data and combining radiological, histological, and pathological information can provide insights into certain biomarkers in the prediction of individual outcomes.

### 3.8. Limitations

There are several limitations of our review. We also did not review or compare CNN prediction of pCR with supervised learning or ML + radiomics [36]. We did not perform meta-analysis of the literature. We did not conduct a statistical analysis pooling the data from multiple studies.

## 4. Conclusions and Future Perspectives

Accurate identification of NAC responder early on in treatment course has the potential to alter therapeutic regimens mid-treatment, improve quality of life and clinical outcomes. It is difficult to predict pCR because the disease is complex and heterogeneous, and there is a large array of longitudinal clinical data that can potentially inform pCR. The multitude of data could be challenging for conventional predictive models. DL prediction of pCR could have a central role to play and could make a positive impact on patient care.

Although in principle DL offers overall advantages compared to supervised ML (such as ML + radiomics) in predicting pCR, studies in the current literature to date generally do not have large enough data to achieve broad generalizability and, thus, the potential of DL in predicting pCR is not yet fully realized. It is also important to vigorously compare different DL models using the same datasets. Large public datasets and/or federated

learning datasets are needed. DL studies that could intelligently integrate multiple types of MRI data (such as, DCE, T2-weighted MRI, fat-water imaging, and diffusion MRI), imaging data at multiple treatment time points, as well as molecular subtypes, demographics, genetic and other data are needed. In addition to predicting pCR, DL models can be used to predict residual cancer burden, progression free survival, risk of recurrence, and overall survival, which are currently understudied. Other DL algorithms beside CNNs should be also explored.

Similar to DL applications to other clinical problems, DL algorithms for prediction of pCR have not yet found widespread clinical applications. Broad adoption of DL in prediction of pCR in clinical settings requires further clinical validation, improved reliability, generalizability, and interpretability, among others. Clinical evaluation needs metrics that are intuitive to physicians and must go beyond measures of technical accuracy to include patient outcomes.

**Author Contributions:** Conceptualization, T.Q.D. and T.M.; Methodology, N.K., R.A., P.H. and T.M.; Formal analysis, N.K., R.A. and T.M.; Data curation, N.K., R.A., P.H. and T.M.; Writing—original draft preparation, N.K. and R.A.; Writing—review and editing, N.K., R.A., P.H., T.M. and T.Q.D. All authors have read and agreed to the published version of the manuscript.

**Funding:** This research received no external funding.

**Institutional Review Board Statement:** No ethics committee approval is needed.

**Informed Consent Statement:** Waived informed consent.

**Data Availability Statement:** Not applicable.

**Conflicts of Interest:** The authors declare no conflict of interest.

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
