# Peer review of "Deep Learning Prediction of Pathologic Complete Response in Breast Cancer Using MRI and Other Clinical Data: A Systematic Review"

_tomography, doi:10.3390/tomography8060232_

Round 1

Reviewer 1 Report

In their systemativ review about "Deep Learning in Breast MRI Cancer Detection and Prediction of Pathologic Complete Response" Khan et al. provide a nice comprehenisve overview about the benefits of AI and deep learning in clinical practice. The authors concluded that deep learning will play a major role in the future practice of radiology. 

The review is well written and easy to understand and read. All the important results are extinsvely discussed in the discussion section. I recommend to accept the study.

Author Response

Reviewer 1:

“In their systematic review about “Deep Learning in Breast MRI Cancer Detection and Prediction of Pathologic Complete Response” Khan et al. provide a nice comprehensive overview about the benefits of AI and deep learning in clinical practice. The authors concluded that deep learning will play a major role in the future practice of radiology.

The review is well written and easy to understand and read. All the important results are extensively discussed in the discussion section. I recommend to accept the study.”

Response: Thank you.

Reviewer 2 Report

All comments are in the attached file, but I must emphasize that there are several studies that are so similar, as for example:

Yu, X., Zhou, Q., Wang, S., & Zhang, Y. D. (2022). A systematic survey of deep learning in breast cancer. International Journal of Intelligent Systems37(1), 152-216.

Author Response

Reviewer 2:

  1. It is very similar to previous paper (Debelee etal. Survey of deep learning in breast cancer mage analysis. Evolving Systems 11, 143-163).

Response: Debelee’s review is a broad review that includes ML in breast cancer image analysis. Our manuscript focuses on more in-depth review of ML on predicting pCR. We believe our study adds to and expands the literature. Thank you for the reference. It is cited.

  1. The reviewer’s raised concern about the words “deep machine learning,” and recommended using “deep learning” or “machine learning” instead.

Response: This was addressed and corrected. It was changed to “deep learning.”

  1. “One of the main problems of using deep learning in medical problems is the fact that the results cannot be interpreted, why do you say it can be done?”

Response:

The following is added.

(b) ML findings are difficult to interpret because there are many features from which ML draws conclusion and ML calculations are complex. There are a few things can be done to improve interpretability. Feature importance and accumulated local effects can be assessed. Additional tools such as by evaluating Shapley values can be employed to explain individual outcomes. For imaging applications, heatmaps can be generated to highlight which regions or features on the images the ML algorithm considers to be important for its calculations.

  1. The reviewer states: “There is no lack of standardization, there a lot of taxonomies about those concepts.”

Response: We removed this statement.

  1. The reviewer requested a reference for our statement that “CNNs, which have 10-30 layers and neurons that share weight since they are connected.”

Response: We removed this statement.

Reviewer 3 Report

Overall:

  • Providing a review of the literature on the topic of deep learning for prediction of pCR in patients receiving NAT is useful and a necessary contribution to the field

  • There are existing recent reviews in the literature on deep learning for breast cancer detection from imaging data that should be considered.

    • Mridha MF, Hamid MA, Monowar MM, Keya AJ, Ohi AQ, Islam MR, Kim JM. A Comprehensive Survey on Deep-Learning-Based Breast Cancer Diagnosis. Cancers (Basel). 2021 Dec 4;13(23):6116. doi: 10.3390/cancers13236116. PMID: 34885225; PMCID: PMC8656730.

    • Rautela, K., Kumar, D. & Kumar, V. A Systematic Review on Breast Cancer Detection Using Deep Learning Techniques. Arch Computat Methods Eng (2022). https://doi.org/10.1007/s11831-022-09744-5

  • There are sections that are not clearly and accurately written in their current form and merit extensive revision. 

  • The authors may consider focusing this review explicitly on DL for pCR prediction as detection is a large topic in itself and has potentially been reviewed (see other reviews above).

  • There is only one figure in the review. It may be helpful to include a visual showing the different approaches of DL methods in BC detection or, perhaps, a general pipeline for BC detection and pCR response (input being the data, a schematic of a general neural network black box, etc).  

Specific comments:

  1. The first reference in the first paragraph of the Introduction is not very clear. Consider this popular reference for cancer statistics instead: This reference does not make sense. Why not reference Siegel, R. L., Miller, K. D., Fuchs, H. E. & Jemal, A. Cancer statistics, 2022. CA: A Cancer Journal for Clinicians 72, 7–33 (2022).

  2. The first paragraph of the Introduction is missing citations, specifically “The American Cancer Society (ACS) recommends yearly screening mammog-raphy for early detection of breast cancer for women, which may begin at age 40. The ACS recommends yearly breast MRI in addition to mammography for women with 20 to 25% or greater lifetime risk.” It is not enough to mention the ACS without the appropriate reference. 

  3. In the Introduction, the authors state that “Across the studies, MRI approximately doubled the cancer yield compared with mam-mography screening alone.” Have the authors considered a discussion on DCIS and the relationship between increased screening and DCIS and BC detection rates? 

  4. Also in the Introduction, the authors state that “ MRI detects the more aggressive/invasive types of breast cancer but has a higher sensitivity than mammography for any type of cancer.” Please cite this statement.  Also, the two statements (before and after the coordinating conjunction) are not opposing. 

  5. Please ensure all statements are cited in the second paragraph of the Introduction (excerpts from this paragraph included below that do not have citations)

    1. Issues such as back-ground parenchymal enhancement that may conceal or simulate a lesion, as well as other perceptual and interpretive errors may be mitigated with the use of artificial intelligence and deep machine learning. Greater accuracy with deep machine learning could avoid unnecessary MRI-guided biopsies. The other advantage of using machine learning/artifi-cial intelligence is increased speed of diagnostic interpretation.”

  6. Please mention the clinical significance of BPE (see Liao GJ, Henze Bancroft LC, Strigel RM, Chitalia RD, Kontos D, Moy L, Partridge SC, Rahbar H. Background parenchymal enhancement on breast MRI: A comprehensive review. J Magn Reson Imaging. 2020 Jan;51(1):43-61. doi: 10.1002/jmri.26762. Epub 2019 Apr 19. PMID: 31004391; PMCID: PMC7207072.)

  7.  On p. 2, the authors state “Machine learning (ML) is an application of artificial intelligence which utilizes an algorithm that computes image features for diagnosis or prediction, allowing the machine to “learn” from experience using statistical techniques.” This is not a general definition of machine learning as the beginning of the sentence suggests it should be. Rewrite generally and cite K.P. Murphy, Machine Learning: A Probabilistic Perspective (MIT Press, 2012).

  8. Continuing from (7), “Under the rubric of ML…”  also isn't a general definition. For an overview of machine learning and deep learning, see Section 4 of Wu C, Lorenzo G, Lima EABF, Hormuth II DA, Slavkova KP, DiCarlo JC, Virostko J, Phillips CM, Patt D, Chung C, Yankeelov TE. Integrating mechanism-based modeling with biomedical imaging to build practical digital twins for clinical oncology. Biophysics Reviews. 2022;3:021304. https://doi.org/10.1063/5.0086789. The references this section cites may be insightful. 

  9. The introductory paragraph on Machine Learning and Deep Learning is hard to follow and does not correctly provide general definitions and, hence, a solid foundation for an introduction to DL in Radiology. Consider rewriting this section general to touch on the following details

    1. A general definition of Machine Learning

    2. A general definition of Deep Learning, falling under the umbrella of ML

    3. A discussion on supervised, self-supervised, and unsupervised methods

    4. How DL methods are trained and what the pros and cons are for each of the three approaches (supervised, self-supervised, unsupervised)

  10. At the top of p. 3, the authors state “One advantage of using neural networks is that the process of feature extraction is merged with decision making so that features are automatically learned, making it possible to detect patterns not seen by humans.” This sentence is convoluted and redundant given that this is by definition what neural networks do. 

  11. Also on p. 3, the authors state that “Models based on deep learning techniques do not necessarily require segmentation of complex shaped regions of interest on images, although in many studies we found radiologists would pre-segment the lesions to decrease processing time and provide ground truth.” Please briefly state what DL models the authors are referring to that do not require complex segmentation of lesions and in what context. 

  12. Figure 1 is not very clear. See Nassif et al. (2022) for a very similar review on Deep Learning for breast cancer detection with a clearer version of your figure in terms of the arrow direction. 

  13. The authors mention radiomics (even in the abstract, suggesting it is important in the manuscript) and juxtapose it with  results of a study on p. 11; however, the authors never define Radiomics in cancer imaging. This review could benefit from a more concrete description of radiomics and how it connects to DL techniques for pCR prediction

    1. See Li et al. Frontiers in Oncology. 2022 (https://www.frontiersin.org/articles/10.3389/fonc.2022.837257/full)

  14. The Prospective/Conclusion is quite general to AI in radiology. It would make the review more comprehensive if the authors specifically focus on the specific topic of the review, summarizing the state of the field, and then touching on general applications of AI in radiology briefly at the end.

  15. In the Prospective/Conclusion in the last paragraph, the authors mention that “Future studies can benefit from institutions working collaboratively to create large datasets suitable for development of reliable deep learning models.” Please consider how Federated Learning may improve collaboration between multiple sites and create more robust models.

Author Response

Reviewer 3:

  1. The first reference in the first paragraph of the Introduction is not very clear. Consider this popular reference for cancer statistics instead: This reference does not make sense. Why not reference Siegel, R. L., Miller, K. D., Fuchs, H. E. & Jemal, A. Cancer statistics, 2022. CA: A Cancer Journal for Clinicians 72, 7–33 (2022).

Response:  The Introduction was rewritten as the focus of the paper is now on prediction of pCR with DL in breast cancer with MRI and clinical data. The previous first reference and the associated statement are no longer in the manuscript.

  1. The first paragraph of the Introduction is missing citations, specifically “The American Cancer Society (ACS) recommends yearly screening mammog-raphy for early detection of breast cancer for women, which may begin at age 40. The ACS recommends yearly breast MRI in addition to mammography for women with 20 to 25% or greater lifetime risk.” It is not enough to mention the ACS without the appropriate reference. 

Response: We removed this statement. We excluded detection and only focused on deep learning of pCR as per recommendation of this reviewer.

  1. In the Introduction, the authors state that “Across the studies, MRI approximately doubled the cancer yield compared with mam-mography screening alone.” Have the authors considered a discussion on DCIS and the relationship between increased screening and DCIS and BC detection rates? 

Response: We removed this statement. We excluded detection and only focused on deep learning of pCR as per recommendation of this reviewer.

  1. Also in the Introduction, the authors state that “ MRI detects the more aggressive/invasive types of breast cancer but has a higher sensitivity than mammography for any type of cancer.” Please cite this statement.  Also, the two statements (before and after the coordinating conjunction) are not opposing. 

Response: We removed this statement. We excluded detection and only focused on deep learning of pCR as per recommendation of this reviewer.

  1. Please ensure all statements are cited in the second paragraph of the Introduction (excerpts from this paragraph included below that do not have citations)
  • Issues such as back-ground parenchymal enhancement that may conceal or simulate a lesion, as well as other perceptual and interpretive errors may be mitigated with the use of artificial intelligence and deep machine learning.
  • Greater accuracy with deep machine learning could avoid unnecessary MRI-guided biopsies. The other advantage of using machine learning/artifi-cial intelligence is increased speed of diagnostic interpretation.”

Response: We removed these statements. We excluded detection and only focused on pCR as per recommendation of this reviewer.

  1. Please mention the clinical significance of BPE (see Liao GJ, Henze Bancroft LC, Strigel RM, Chitalia RD, Kontos D, Moy L, Partridge SC, Rahbar H. Background parenchymal enhancement on breast MRI: A comprehensive review. J Magn Reson Imaging. 2020 Jan;51(1):43-61. doi: 10.1002/jmri.26762. Epub 2019 Apr 19. PMID: 31004391; PMCID: PMC7207072.)

Response: We added the following statement regarding BPE and cited the above reference Liao et al:

background parenchymal enhancement by MRI is informative of cancer risk, recurrence, and outcomes but has not yet been adequately explored for predicting pCR [29].

  1. On p. 2, the authors state “Machine learning (ML) is an application of artificial intelligence which utilizes an algorithm that computes image features for diagnosis or prediction, allowing the machine to “learn” from experience using statistical techniques.” This is not a general definition of machine learning as the beginning of the sentence suggests it should be. Rewrite generally and cite P. Murphy, Machine Learning: A Probabilistic Perspective (MIT Press, 2012).

Response: It is rewritten. Thank you for the reference. It is cited.

  1. Continuing from (7), “Under the rubric of ML…”  also isn't a general definition. For an overview of machine learning and deep learning, see Section 4 of Wu C, Lorenzo G, Lima EABF, Hormuth II DA, Slavkova KP, DiCarlo JC, Virostko J, Phillips CM, Patt D, Chung C, Yankeelov TE. Integrating mechanism-based modeling with biomedical imaging to build practical digital twins for clinical oncology. Biophysics Reviews. 2022;3:021304. https://doi.org/10.1063/5.0086789. The references this section cites may be insightful.

Response: This section was rewritten, and we removed this statement.

  1. The introductory paragraph on Machine Learning and Deep Learning is hard to follow and does not correctly provide general definitions and, hence, a solid foundation for an introduction to DL in Radiology. Consider rewriting this section general to touch on the following details
    1. A general definition of Machine Learning
    2. A general definition of Deep Learning, falling under the umbrella of ML
    3. A discussion on supervised, self-supervised, and unsupervised methods
    4. How DL methods are trained and what the pros and cons are for each of the three approaches (supervised, self-supervised, unsupervised)

Response: That section is revised as follow:

Machine learning (ML) is increasingly being used in radiology and medicine [6-8]. In contrast to conventional analysis methods which need to specify the relationships amongst data elements to outcomes, ML employs computer algorithms to identify relationships amongst different data elements to inform outcomes without the need to specify such relationships a priori. ML can outperform human experts in performing many tasks in medicine [9]. In addition to approximating physician skills, ML can also detect novel relationships not readily apparent to human perception, especially in large complex datasets. There are multiple types of learning methods (supervised, self-supervised, and unsupervised), each having its own advantages and disadvantages. Supervised learning learns from labeled data to perform tasks such as prediction and classification of data. A comparatively small data set is needed but the disadvantage is that labeled datasets for a specific task needs to be provided. Self-supervised learning exploits unlabeled data to yield labels. This eliminates the need for manually labeling data, which is a tedious process, but larger datasets are needed. Supervisory signals provide feedback to train the network, without requiring a labeled dataset. Unsupervised learning needs no corresponding classification or label, and the algorithm finds underlying patterns with each dataset. The disadvantage is that is it requires large datasets and interpretation with respect to clinical findings could be challenging. One of the most popular unsupervised learning, deep learning (DL) methods [10] used on images is the Convolutional Neural Networks (CNN) [11]. A CNN is well suited for image recognition and tasks that involve the processing of pixel data. CNN is particularly suitable for computer vision tasks and for applications where object recognition such as facial recognition and radiological images. CNN methods do not require radiologists to contour the tumor on images.

  1. At the top of p. 3, the authors state “One advantage of using neural networks is that the process of feature extraction is merged with decision making so that features are automatically learned, making it possible to detect patterns not seen by humans.” This sentence is convoluted and redundant given that this is by definition what neural networks do. 

Response: We removed this statement to avoid confusion.

  1. Also on p. 3, the authors state that “Models based on deep learning techniques do not necessarily require segmentation of complex shaped regions of interest on images, although in many studies we found radiologists would pre-segment the lesions to decrease processing time and provide ground truth.” Please briefly state what DL models the authors are referring to that do not require complex segmentation of lesions and in what context.

Response: We removed this statement and refocused the paper on deep learning prediction of pCR.

      12.  Figure 1 is not very clear. See Nassif et al. (2022) for a very similar review on Deep Learning for breast cancer detection with a clearer version of your figure in terms of the arrow direction. 

Response: Thank you. A new flowsheet is provided.

     13. The authors mention radiomics (even in the abstract, suggesting it is important in the manuscript) and juxtapose it with results of a study on p. 11; however, the authors never define Radiomics in cancer imaging. This review could benefit from a more concrete description of radiomics and how it connects to DL techniques for pCR prediction

See Li et al. Frontiers in Oncology. 2022

Response: We agree. In this revised paper, we have removed mention of radiomics and focus on DL of pCR prediction. Thank you.

  1. The Prospective/Conclusion is quite general to AI in radiology. It would make the review more comprehensive if the authors specifically focus on the specific topic of the review, summarizing the state of the field, and then touching on general applications of AI in radiology briefly at the end

Response: The revision is provided below:

Accurate identification of NAC responder early on has the potential to alter therapeutic regimens mid-treatment, improve quality of life and clinical outcomes. It is difficult to predict pCR because the disease is heterogeneous, and there is a large array of longitudinal clinical data that can potentially inform of pCR outcome. The multitude of data could be challenging for conventional predictive models. DL prediction of pCR has a central role to play and could make a positive impact.

Although in principle DL offers overall advantages compared to supervised ML (such as ML + radiomics) in predicting pCR, studies in the current literature do not appear to have large enough data to achieve broad generalizability and, thus, the full potential of DL in predicting pCR is not yet fully realized. Large public datasets and/or federated learning datasets are needed.  DL studies that properly or intelligently integrate multiple types of MRI data (such as, DCE, T2-weighted MRI, fat-water imaging, and diffusion MRI), imaging data at multiple treatment time points, as well as molecular subtypes, demographics, genetic and other data are needed. In addition to predicting pCR, DL models can be used to predict residual cancer burden, progression free survival, risk of recurrence, and overall survival, which is currently understudied. Other DL algorithms beside CNNs should be also explored.

Similar to DL applications to other clinical problems, DL algorithms for prediction of pCR have not yet found widespread clinical applications. Broad adoption of DL in prediction of pCR in clinical settings requires further clinical validation, improved reliability, generalizability, and interpretability, among others. Clinical evaluation needs metrics that are intuitive to physicians and must go beyond measures of technical accuracy to include patient outcomes.

  1. In the Prospective/Conclusion in the last paragraph, the authors mention that “Future studies can benefit from institutions working collaboratively to create large datasets suitable for development of reliable deep learning models.” Please consider how Federated Learning may improve collaboration between multiple sites and create more robust models

Response: We added the following to the manuscript:

To overcome limited dataset, federated learning or collaborative learning can be employed in which a machine learning algorithm is trained on multiple local datasets without data sharing or data exchange, thus preserving data privacy and security.

Reviewer 3 Overall:

  1. Providing a review of the literature on the topic of deep learning for prediction of pCR in patients receiving NAT is useful and a necessary contribution to the field

Response: Thank you.

  1. There are existing recent reviews in the literature on deep learning for breast cancer detection from imaging data that should be considered.
    1. Mridha MF, Hamid MA, Monowar MM, Keya AJ, Ohi AQ, Islam MR, Kim JM. A Comprehensive Survey on Deep-Learning-Based Breast Cancer Diagnosis. Cancers (Basel). 2021 Dec 4;13(23):6116. doi: 10.3390/cancers13236116. PMID: 34885225; PMCID: PMC8656730.
    2. Rautela, K., Kumar, D. & Kumar, V. A Systematic Review on Breast Cancer Detection Using Deep Learning Techniques. Arch Computat Methods Eng (2022). https://doi.org/10.1007/s11831-022-09744-5

Response:  We excluded detection and only focused on pCR as per recommendation of this reviewer and therefore these references on detection are no longer necessary.

      3. There are sections that are not clearly and accurately written in their current form and merit extensive revision. 

Response: We carefully revised and edited the entire manuscript. Thank you.

     4. The authors may consider focusing this review explicitly on DL for pCR prediction as detection is a large topic in itself and has potentially been reviewed (see other reviews above).

Response: We agree and removed ML detection and focused on DL for pCR prediction.

      5. There is only one figure in the review. It may be helpful to include a visual showing the different approaches of DL methods in BC detection or, perhaps, a general pipeline for BC detection and pCR response (input being the data, a schematic of a general neural network black box, etc).  

Response: We are including an additional figure as recommended.

Round 2

Reviewer 2 Report

"There are many review papers on using ML analysis of MRI data to predict pCR in breast cancer [13, 14]. However, most papers review ML methods on are supervised ML, such as support vector machine, random forest, decision tree, extreme gradient besting, Boosted tree, Bayesian methods, among others [13, 14]."

It is boosting, not besting. You say in the response that [14] is not a pCR prediction survey, but in this sentence you say the contrary. You should give more compelling reasons to show that the papers are different.

Author Response

"There are many review papers on using ML analysis of MRI data to predict pCR in breast cancer [13, 14]. However, most papers review ML methods on are supervised ML, such as support vector machine, random forest, decision tree, extreme gradient besting, Boosted tree, Bayesian methods, among others [13, 14]."

It is boosting, not besting.

Response: corrected

You say in the response that [14] is not a pCR prediction survey, but in this sentence you say the contrary. You should give more compelling reasons to show that the papers are different.

Response: We apologize for the error in the response. It is revised to make more compelling that our paper is different:

There are many review papers on using ML analysis of MRI data to predict pCR in breast cancer (see reviews [13, 14]) but most were of supervised ML, such as support vector machine, random forest, decision tree, extreme gradient boosting, Boosted tree, Bayesian methods, among others (see reviews [13, 14]). Supervised ML methods use extracted image features (i.e., tumor volume, diameter, radiomic features), that usually require expert radiologists to contour the tumors and/or identify features to train the ML algorithms. By contrast, reviews that focus on DL methods (such as CNN) in which whole breast MRI images are used without manual contouring or annotation of the tumor to predict pCR is limited [13-15].

We thus took on a review that focuses on DL (i.e., CNN) which allows whole-breast MRI images without annotation or tumor segmentation to predict pCR in breast cancer. Our review differed from prior related reviews in that we compared image types used (i.e., DCE or post contrast), whether pre-training, transfer learning or data augmentation was used, whether CNN models include molecular subtypes, multiple treatment time points, multi-institutional data, as well as whether saliency maps (heatmaps) were provided. We also compared performance metrics across different DL studies that predict pCR.

Reviewer 3 Report

The authors have addressed all of my points, thank you. 

My only remaining concern is that Figure 2(B) is a bit hard to read. Please make the blocks and fonts within them bigger. Also "time point #1" and "time point #2" are a bit close together, so it's not obvious that they are labelling the blocks beneath them. Maybe abbreviate to t_1 and t_2 and explain these labels in the caption?

Author Response

My only remaining concern is that Figure 2(B) is a bit hard to read. Please make the blocks and fonts within them bigger. Also "time point #1" and "time point #2" are a bit close together, so it's not obvious that they are labelling the blocks beneath them. Maybe abbreviate to t_1 and t_2 and explain these labels in the caption?

Response: We apologize for the error. It is fixed.